# A Psychometric Study of the Arabic Version of the “Searching for Hardships and Obstacles to Shots (SHOT)” Instrument for Use in Saudi Arabia

**DOI:** 10.3390/vaccines12040391

**Published:** 2024-04-09

**Authors:** Fatimah Hobani, Manal Alharbi

**Affiliations:** 1Collage of Nursing, King Saud University, Riyadh 11451, Saudi Arabia; 2Primary Health Sector, Jizan Health Cluster, Ministry of Health, Jizan 84421, Saudi Arabia; 3Department of Maternal and Child Health, Collage of Nursing, King Saud University, Riyadh 11451, Saudi Arabia; maalwahbi@ksu.edu.sa

**Keywords:** children immunization, barriers, obstacles, vaccines, Saudi Arabia, psychometric tools for vaccine hesitancies

## Abstract

Vaccines are considered one of the top 10 public health achievements of the 20th century and the most cost-effective public health intervention to overcome diseases and disease-associated mortality. This study translated the “Searching for Hardships and Obstacles to Shots” (SHOT) instrument from English to Arabic and conducted a psychometric evaluation of the Arabic version to measure parental barriers to childhood immunization. The cross-sectional study utilized multistage cluster random sampling to recruit parents visiting 70 primary health centers in Jizan. Scale translation and cultural adaptation were used to translate the SHOT survey into Arabic. The survey revealed that the best-factor model was a one-factor solution for “barriers to child immunization.” The first principal component explained the highest variance (56.22%), and subsequent components explained decreasing percentages of variance. The third principal component explained the decreased variance (4.61%), and subsequent components explained the decreasing percentages of variance. The overall reliability (determined by Cronbach’s alpha) was 0.96. The strong internal consistency of the Arabic version of the SHOT instrument (as indicated by the high Cronbach’s alpha coefficients) indicates that researchers and practitioners can confidently use this scale to measure parents’ attitudes toward and perceptions of vaccinations. Furthermore, the study results will help policymakers develop programs or interventional initiatives to overcome these barriers.

## 1. Introduction

Vaccines are considered one of the top 10 public health achievements of the 20th century [1] and are the most cost-effective public health intervention for overcoming diseases and disease-associated mortality [2]. The Centers for Disease Control and Prevention (CDC) recommends approximately 16 vaccines for children between birth and the age of 18 months [3]. The World Health Organization (WHO) recently estimated that the percentage of vaccinated children decreased from 86% in 2019 to 83% in 2021 globally [4]. In 2018, worldwide, approximately 5.3 million children aged five years were exposed to infectious diseases, leading to an estimated 700,000 deaths [5]. The CDC recommends the following immunization series for children before 2 years of age: hepatitis B, rotavirus, diphtheria–tetanus–acellular pertussis, Haemophilus influenza type B, pneumococcal conjugate vaccine, inactivated poliovirus, measles–mumps–rubella, varicella, hepatitis A, and meningococcal [3].

Parental misconceptions of immunization are a major obstacle to immunizing children [6]. Additionally, timely vaccination is hindered by parental hesitancy to vaccinate their children because of concerns about vaccine safety or beliefs that vaccines are not necessary [7]. The COVID-19 pandemic had an impact on the healthcare system, including immunization programs [8]. Routine childhood vaccination coverage rates have declined in many countries during the COVID-19 pandemic [9], with global coverage estimated at 76.7% [10]. A systematic review of the 2021 results suggests a decrease in vaccination coverage and the overall number of vaccines provided, leading to children missing out on their vaccine doses [8].

Previous studies have identified several barriers to parents choosing to vaccinate their children; parents’ choice to reject or postpone childhood vaccinations is a key factor in reducing immunization coverage; such decisions are based on complex beliefs [11]. Vaccine refusal occurs in many countries, and the incidence is gradually increasing. Decreasing vaccination rates have led to a significant increase in the incidence of vaccine-preventable diseases, such as measles, chickenpox, and hepatitis A [12]. In 1998, Andrew Wakefield published a study on the connection between the measles, mumps, and rubella vaccine and autism. The study was retracted, but it had already caused substantial damage [1]. Skepticism about vaccines increased after this publication, which was made possible by increasingly negative media coverage of the MMR vaccine [13]. This is an example of anti-vaccine movements that aim to disrupt vaccine programs because of the false link between autism and immunization [14]. Public health administrators must understand parents’ perspectives on accepting or rejecting recommended immunizations for their children to improve vaccination coverage and implement effective campaigns. This understanding is also vital for effective communication about vaccination between parents and healthcare providers [11].

The recent deterioration in vaccine uptake is attributed to multiple factors, but the two main ones are parents’ low confidence in vaccines and their perception that the risks associated with vaccines are high [15]. Parents who delay vaccination or refuse to vaccinate their children contribute considerably to lower immunization rates; this decision is influenced by complicated beliefs [16]. Children rely on their parents for their health decisions [17], and parents’ refusal or delay of vaccination may impact their children’s health outcomes [18].

Vaccine acceptance has become an emerging global problem [19]. Rumors have a negative impact on parents’ attitudes and perceptions regarding childhood immunization, which leads to mistrust and contributes to vaccine hesitancy [20]. Also, the internet has been significantly associated with a negative perception of vaccine risks [21]. On the other hand, a review found that social and cultural influences influenced parents’ choices regarding their children’s vaccinations. Advice from peers, family members, and friends is a source of information and social influence in vaccination decision-making [22].

In terms of vaccination non-adherence, a cross-sectional study conducted in Saudi Arabia revealed the percentage of non-adherence with children immunization and delayed immunizations (59.1%) compared with 40.9% who received the vaccines on time [23]. However, few studies have explored parents’ perceptions of and attitudes toward child immunization [24]. Parents accept the importance of childhood immunization when children have been incompletely vaccinated; the most hesitant and delayed vaccination appointments are with parents who are highly educated [25,26]. In addition, this negatively impacts the immunization status among children.

Parents’ lack of commitment to vaccinate their children contributes greatly to the spread of infectious diseases; this leads to an increased incidence of illness in children, and their treatment places a burden on public health resources [6] and leads to a rise in the incidence of infectious diseases [27]. For instance, research has found that in several countries, parents are fearful of child immunization, exacerbating the spread of infectious diseases [28]. Such concerns about vaccinations are widespread and exacerbate health system-related barriers to immunization [7]. Recently, health systems have faced increasing challenges, with more parents choosing not to vaccinate their children. This vaccine refusal persists despite the extensive research supporting the effectiveness and safety of vaccines, and it has crucial implications for community health worldwide [2].

In terms of the social and political determinants of vaccine hesitancy, several factors were identified [29]. Multiple factors impact immunization rates, including parents’ health behaviors; parents who adopt healthy behaviors are more likely to immunize their children than parents who do not [30]. Social determinants, such as a younger parent’s age, parental education level, family income, lack of health insurance, lack of access to periodic primary health care, and the cost of vaccinations, also affect vaccination rates [30]. Political determinants revolve around confidence in the government’s technical and organizational skills to deal with infectious disease outbreaks and trust in medical organizations’ abilities to predict the adoption of recommended protective measures [29]. However, public compliance with vaccination plans in health crises requires developing social and institutional trust [29].

Several instruments to measure vaccine hesitancy were identified [19,27,31]. Previous studies identified problems with interpreting vaccine hesitancy, especially in vaccine shortages, and used a Likert scale that does not resonate across diverse cultural settings [19]. In another study measuring vaccine five hesitancy, all liability has been highlighted as barriers to vaccination uptake among parents to varying degrees for the 5Cs; it is a scale used to assess the psychological antecedents (antecedents means the cause or determinant that relates to vaccination behavior) of vaccination on measuring vaccine hesitancy, which are confidence, complacency, constraints calculation, and collective responsibility [32]. Also, the multidimensional vaccine hesitancy scale was valid for measuring this issue and is suitable for clinical practice and research analyzing vaccination behaviors and intentions [27]. In addition, trust in the system and social compliance among antecedents of vaccine acceptance allowed for a better understanding of the transition from rejection to hesitation and acceptance [33].

The “Searching for Hardships and Obstacles to Shots” (SHOT) instrument was designed to evaluate the validity and reliability of the Arabic-translated version to measure parental barriers to childhood immunizations; this study translated this instrument into Arabic and assessed its psychometric properties. To evaluate the validity of the translated SHOTS survey, it was evaluated in an Arab cultural context. The study results may enhance a better understanding of barriers to childhood immunization.

## 2. Materials and Methods

### 2.1. Study Design

The cross-sectional study utilized multistage cluster random sampling to recruit parents visiting 70 primary health centers in Jizan. Scale translation and cultural adaptation were used to translate the SHOT survey into Arabic.

### 2.2. Study Sample and Setting

This two-phase, cross-sectional study was conducted in primary healthcare centers (PHCs) in the Jizan region of Saudi Arabia. A multistage cluster random sampling technique was applied. In the first phase, the scale was culturally adapted and translated into Arabic. In the second phase, 600 Saudi parents were recruited via convenience sampling to complete the survey. The Saudi National Immunization Schedule begins at birth and continues until children are school-aged. Therefore, this study included Saudi parents with children aged 6 years or younger who visited PHCs. The following exclusion criteria were applied: non-Saudi nationality, having children aged over 6 years, and an unwillingness to participate in the study.

In general, the estimated total population size in Saudi Arabia is about 32,175,224 in 2022 and, specifically, in Gizan city, 1,404,997. According to the Saudi Annual Statistic Book, the total number of PHCCs in Saudi Arabia in 2022 reach approximately 2390 [31]. In the city of Jizan (located in the southwestern corner of Saudi Arabia), a list of primary health care centers affiliated with the Ministry of Health was taken. Approximately 75 participants were randomly selected from each sector in Jizan. Jizan city health cluster has approximately 170 PHCCs spread over seven sectors; below each sector, there are many PHCCs: Central 16, Farasan Island 4, Southern 38, Middle 32, Western 30, and the Eastern sector incorporates Mountain 13 PHC and Bani-Malik 12 PHC, and North 25 PHCs.

Cluster random sampling was employed using a proportional allocation of participants to collect data from all sectors. Specifically, 10 PHCs were selected from each cluster, and 8 participants were selected from each PHC. A list of the 170 health centers was obtained from the Ministry of Health. Web-based (https://www.random.org, accessed on 4 May 2023) was used to randomly select 10 PHCs from each sector. A total of 605 links were sent, and 602 responses were collected; thus, the response rate was 99.5%. This rate was high because the researchers followed the process of data collection with the data collectors daily. There were no missing data [34].

### 2.3. Data Collection Procedure

Data were collected from June to August 2023 using an online questionnaire. Parents visiting PHCs were invited if they met the eligibility criteria and were approved. The sample size was calculated with an online a priori sample size calculator for a structural equation model (SEM) (https://www.danielsoper.com/statcalc/calculator.aspx?id=89, accessed on 6 March 2023) with an alpha of 0.05, a power of 0.80, 23 observed variables, and 3 latent variables. A medium effect size of 0.3 was used. The minimum sample size to detect an effect was 119, the minimum sample size for the model structure was 589, and the recommended minimum sample size was 589. Therefore, a total of 600 participants were recruited [35].

### 2.4. Ethical Concerns

Before data collection, permission was requested from the institutional review board of King Saud University and the Saudi Ministry of Health to access data from the PHCs in Jizan, Saudi Arabia. A signed agreement with the authors of the SHOT tool was obtained via email to use, translate, and adapt the tool. Participants were informed about the study’s purpose, procedures, risks, and benefits. Informed consent was signed when they agreed to participate by clicking on the provided link while answering the questionnaire. Participation in the study was voluntary, and participants could choose to withdraw at any time. Participant names were substituted with codes for data analysis. Maintaining data security and confidentiality was a top priority throughout the research process.

### 2.5. Instrumentation

This study utilized the SHOT questionnaire, which was originally developed by Victoria P. Niederhauser. The scale is based on the theory of reasoned action (TRA) [36] and examines parent-related barriers to childhood immunization to identify obstacles to vaccinations. The survey was tested on parents of children aged 8 years or younger. Each item is rated on a five-point Likert scale, representing the degree to which the parents view the item as a problem (from “zero = not a problem” to a “5 = very big problem”). The self-administered questionnaire contains 23 items across three subscales: the access to shots subscale (items 1–3, 0–48 points), the concerns about shots subscale (items 13–18, 0–24 points), and the importance of shots subscale (items 19–23, 0–20 points). The total score on the questionnaire is obtained by summing the points; the total score ranges from 0 to 92. A higher score indicates that parents face a more difficult combination of elements when vaccinating their children. The overall internal consistency reliability of the SHOT survey was determined by Cronbach’s alpha, which was calculated to be 0.93. The Cronbach’s alpha values of the subscales were as follows: access to shots, 0.92; concerns about shots, 0.88; and importance of shots, 0.86. The initial testing of the SHOT instrument revealed good viability. The SHOT instrument has not been previously tested in the Saudi population. Thus, psychometric analysis of the instrument (which is essential for adapting, translating, and evaluating scale reliability) was required because the instrument had not been tested in the Arab population previously (convenience sampling).

### 2.6. Data Analysis

The data were analyzed using the Statistical Package for Social Sciences (SPSS) version 29 software. Descriptive statistics were used to present the demographic characteristics of the study participants, and the scale’s internal consistency reliability was assessed by calculating Cronbach’s alpha. Normality was examined, as was the skewness and kurtosis of each study variable, to determine whether the distribution was normal. According to Kline (2016), data are not normally distributed if the skewness index (SI) is greater than or equal to 3 and the kurtosis index (KI) is greater than or equal to 10 [37]. Factor analysis was performed because the instrument had not been previously applied in an Arabic cultural context. To evaluate construct validity, exploratory factor analysis was conducted [38] through the principal axis analysis method, which defines acceptable factor-loading values as those higher than 0.30 [39]. The Kaiser–Meyer–Olkin (KMO) test for sampling adequacy and Bartlett’s test of sphericity were evaluated before performing the exploratory factor analysis [38].

## 3. Results

### 3.1. Cultural Adaptation and Scale Translation

In the translation process, the conceptualization and operationalization of health-related concepts may differ across cultures. These differences underscore the importance of considering conceptual equivalence when adapting and translating survey tools. In this study, the SHOT survey was adapted and translated. A built-in method of translating the scale was used after adapting it to the culture and language of the population [40]. To avoid cultural and language barriers, a five-stage integrated approach was applied; these five stages were as follows: choosing measures of cultural adaptation and translation, assessing conceptual equivalence, forward translation, back translation (optional), and a pre-testing phase to assess cultural relevance [40].

Bicultural and bilingual professionals with backgrounds in Arabic and English languages and cultures read all the items separately, considering recognizable components and their importance to Arab culture; they are bilingual in the Arabic and English languages, and they are knowledgeable about vaccine hesitancy. They evaluated each component using a 10-point scale (1 = not at all understandable; 10 = fully understandable; 1 = inappropriate; and 10 = very relevant). Subsequently, they met with a team of healthcare experts to clarify the meaning of the items, ensure the scale’s appropriateness in the Saudi Arabian cultural context, and make any necessary adjustments. During this meeting, the evaluators and the team of experts decided that none of the elements needed amendment. The items were evaluated as easy to understand and culturally relevant (rated above 5). The evaluators and experts also assessed other elements of the scale to ensure that the language was clear and free from ambiguity, the words were easy to understand, the questions were not difficult, and the content was culturally relevant. They also approximated the required time for the participants to complete the questionnaire. For face validity, 10 experts (five nurses and five family medicine physicians) assessed the survey to ensure that the items were clear, understandable, answerable, and free from ambiguity.

The final list of scale content was forward-translated by two translators. One of the translators was a bilingual healthcare professional with a background in the barriers to children immunization, and the other was a healthcare professional who was bilingual and bicultural [40]. They independently completed the initial translation of all the elements included in the final list created during the last stage. Before translation, they were reminded to preserve the meaning of the items in the original scale, incorporate appropriate cultural expressions, and use terms that were simple, clear, easy to understand, and recognizable. After translation, the translators discussed the difficulty of translating the items, the adequacy of their understanding, and the cultural appropriateness of the translations. If any inconsistencies or conflicts arose, the reasons were identified. The researchers and translators suggested alternative Arabic wording for the material until both approved and accepted it without any disagreement. For instance, when experts suggest the translated words, others in the group also suggest other synonyms until they reach suitable wording. Ultimately, a consensus was reached, and the scale translation was accepted. Back translation (i.e., translating a scale from its target language back to the original language) is considered optional [40]; thus, this step was not performed in this study.

Finally, two groups of participants were recruited for the pre-testing phase (pilot study) to assess the cultural relevance. The first group of five bilingual, bicultural nurses and physicians participated in cognitive interview sessions, and the second group of five monolingual nurses and physicians participated in the concept of interest. The feedback from the first group of participants was used to assess the quality of the translated version; the comments from the second group were important in determining the understanding and cultural significance of the items translated into Arabic. Participants completed a checklist containing the items in both English and Arabic and rated them on a 10-point scale (1 = not at all understandable, 10 = entirely understandable; 1 = inappropriate, 10 = very relevant; 0 = not obvious word meaning and 10 = very obvious word meaning). After completing the assessments, the participants discussed the items that were unclear or difficult to understand, as well as the possible reasons for the difficulty and ways to make the items more transparent; furthermore, they identified whether the content or wording of any items was culturally irrelevant and provided suggestions for enhancing cultural relevance [40]. The second group of monolingual participants completed the scale in Arabic independently. The bilingual researchers met and discussed the clarity and cultural significance of each item.

Following an established method for adapting surveys to the culture and language of the target population [40], a list of items in Arabic was completed and subjected to further testing of the psychometric properties. The reliability of the scale (internal consistency) was verified through a pilot study/psychometric analysis involving a convenience sample of 600 Saudi parents with children between birth and school age; the participants were recruited from the PHCs in Jizan and completed a self-administered questionnaire. This stage was necessary because the SHOT scale had not been used in an Arab cultural context before this study. Therefore, a pre-test of the survey was conducted in the target population. Pre-testing helps eliminate poorly worded items and ensures that the revised wording is fully understood, thus reducing misunderstanding and measurement error [41]. The pilot study involving 600 parents also assessed subjective norms, behavioral intention, and perceived behavioral control.

### 3.2. Content Validity

The content validity index (CVI) was calculated using the content validity index (CVI) and was measured as the sum of the selected numbers of experts divided by the number of experts. Item adjustments will be necessary if the item is evaluated as not easy to understand (i.e., a comprehension index less than or equal to five) or inappropriate (i.e., CVI less than 78%). Items are deleted if they are deemed unsuitable to Saudi culture. Therefore, ten experts in children’s immunization reviewed the survey for content validity. The value was 95%, indicating that the scale is explicit language, easy to understand, free from difficulty, and culturally relevant. Consequently, no modifications were recommended. During translation, the experts suggested no cultural conflict or cultural sensitivity found in the current tool.

### 3.3. Back-Translated English Version

According to Sidani et al., 2010, back translation is unnecessary in the proposed process of adapting and translating an instrument and is considered optional [40]. This is because two translated versions of the same instrument (one with back translation and the other without back translation) had identical psychological properties [42]. Therefore, this step was not performed.

### 3.4. Testing of the SHOT Tool among the Community

#### 3.4.1. Background Characteristics of the Study Participants

The demographic characteristics of the sample are described in Table 1. The majority of participants belonged to the central sector, 31.2% (n = 188). Most of the study participants were female, 83.7% (n = 504), and 16.3% (n = 98) were male. Of the parents declaring that their children received vaccination, 95.7% (n = 576), and 4.3% (n = 26) of children did not receive their immunization. Furthermore, 93.9% (n = 565) reported that their children receive vaccinations regularly according to the MOH schedule and 6.1% (n = 37) reported as being not regularly received. Also, 92.5% of participants were married (n = 557), 2.3% (n = 14) were widows, and only 5.1% (n = 31) were divorced. The descriptive statistics for each variable are reported in Table 2.

Table 3 provides the results of the survey, in which the respondents rated their agreement or concern levels regarding different statements related to child vaccinations. The questions’ 5-point Likert scales were coded from 0 to 4. The table presents the mean (average) and standard deviation (SD) for each statement on the scale. The mean values reflect the respondents’ average levels of agreement or concern for each statement. For instance, statements such as “My child was sick and could not get his/her shots” and “I worry my child might get sick from the shot” had higher mean scores (1.11 and 1.03, respectively), indicating stronger agreement or higher levels of concern among respondents regarding these issues. Statements such as “I don’t believe in getting kids shots” and “I don’t think kids’ shots are important” had lower mean scores (0.60 and 0.58, respectively), suggesting lower agreement or concern regarding these issues among respondents.

The SD represents the spread of responses around the mean. Higher SD values indicate a greater variability in the respondents’ opinions of or concerns about a particular statement. For example, statements such as “My child was sick and could not get his/her shots” and “I worry about the number of shots my child gets at one time” had relatively high SD values (1.43 and 1.32, respectively), indicating a high variability in respondents’ opinions about these concerns. Statements such as “I don’t believe in getting kids shots” and “I don’t think kids’ shots are important” had relatively low SD values (1.23 and 1.20, respectively), suggesting a low variability in the responses to these statements.

Overall, the mean scores and SDs provide insights into the respondents’ levels of agreement with or concerns about various statements related to child vaccinations, as well as the degree of variability of these opinions in the surveyed population.

#### 3.4.2. Scale Reliability

The scale’s internal consistency reliability was determined by calculating Cronbach’s alpha (a), and the following values were obtained: access to shots, a = 0.94; concerns about shots, a = 0.927; importance of shots, a = 0.90; and overall SHOT scale, a = 0.96. These Cronbach’s alpha coefficients demonstrate that the scale adequately assessed access to vaccinations, concerns about vaccinations, the importance of vaccinations, and the overall SHOT scale, exhibiting strong internal consistency between the items used to measure these constructs. High Cronbach’s alpha values indicate that the items within the scale are strongly correlated with each other, implying a high degree of reliability or consistency in measuring the respective constructs related to vaccine access, concerns, and importance, as well as the overall vaccination-related attitudes and perceptions of the respondents.

#### 3.4.3. Construct Validity Assessment: Exploratory Factor Analysis

Both exploratory factor analysis and item analysis were performed to assess the validity and reliability (internal consistency) of the Arabic version of the SHOT instrument. The main purpose of factor analysis was item reduction and validity and reliability assessment of the Arabic version. The results showed that all 23 items were loaded on one item factor; therefore, all 23 items were considered barriers to child immunization. No items needed to be deleted. All 23 items represented 67% of the barriers, and the remaining percentage was related to other factors. All items loaded on factor one were equal to over 0.5, which is considered as very high representativeness.

Table 4 provides the statistics related to the item analysis of the SHOT scale. These statistics help to evaluate the performance and reliability of individual items in the scale. The scale mean indicates how the mean score of the entire scale would change if a specific item were removed. Higher differences suggest that the item significantly impacts the overall mean score.

The scale variance indicates how the variance of the scale scores would change if a specific item was removed. Larger changes suggest that the item contributes significantly to the variability within the scale. The corrected item-total correlation statistic reflects how well an individual item correlates with the overall scale score after removing that item. Higher correlation values indicate that the item is more closely related to the overall scale score.

Cronbach’s alpha indicates the reliability of the scale (internal consistency) if a specific item is removed. A decrease in Cronbach’s alpha after removing an item suggests that the item contributes to the scale’s overall reliability. For instance, items with higher corrected item-total correlations (close to 1) and higher Cronbach’s alpha values, if deleted (closer to the overall alpha), are considered to have strong consistency and a high contribution to the scale’s reliability. Items with larger changes in scale mean or variance, if deleted, may have a more significant impact on the overall scale scores and variability. These statistics help to evaluate the performance and contribution of individual items to the overall scale, assisting in identifying items that may require revision, deletion, or further investigation for scale refinement and improvement.

The results provided are related to the KMO test for sampling adequacy and Bartlett’s test of sphericity. These tests are often conducted before performing factor analysis, including principal component analysis (PCA), to assess whether the data are suitable for such analyses. The KMO statistic measures the adequacy of the data for the application of techniques such as PCA and factor analysis. The KMO value ranges from 0 to 1; higher values (closer to 1) indicate that the dataset is more suitable for factor analysis. In this case, the KMO value was very high (0.966), indicating that the variables in the dataset were well-suited for factor analysis. Values above 0.6 or 0.7 are generally considered acceptable for factor analysis. Bartlett’s test of sphericity assesses whether the correlation matrix between variables is an identity matrix, indicating that the variables are unrelated and unsuitable for structure detection. The test statistic approximates the chi-square statistic and assesses whether correlations between variables are sufficiently large for factor analysis to be useful. The “Approx. Chi-Square” value was 10,905.20 with 253 degrees of freedom and a significance level (Sig.) of <0.00 (i.e., very close to zero), indicating that the correlations between variables were significantly different from those of an identity matrix. A significant result (i.e., a *p*-value less than a chosen significance level, often 0.05) of Bartlett’s test suggests that the variables have a sufficient correlation for factor analysis.

The high KMO value (0.96) and the significant result for Bartlett’s test (Sig. = <0.00) indicated that the dataset was highly suitable for factor analysis or related techniques. The variables in the dataset were correlated enough to proceed with techniques such as PCA or factor analysis, suggesting that these techniques could be applied confidently to derive meaningful insights or reduce dimensionality while preserving important information in the dataset.

Table 5 provides information about the communalities obtained from the PCA. Communalities represent the proportion of variance in each observed variable that is accounted for or explained by the extracted components. In PCA, variables are transformed into a smaller set of linearly uncorrelated variables (principal components). Communalities indicate how well each original variable is represented by these components.

Initial communalities are the communalities before the PCA, for which each variable’s variance is set to 1.00 (100%). Extraction communalities show the proportion of variance in each variable that is accounted for by the principal components extracted through PCA. For instance, the first item (“I didn’t know when my child needed to get his/her shots”) had an initial communality of 1.000, indicating that all its variance was accounted for in the original data.

After the PCA, the extraction communality for the same item was 0.63, suggesting that the principal components extracted accounted for 63.6% of the variance in this item. Higher extraction communalities (closer to 1) indicate that the principal components captured a larger portion of the variance in the original items. Lower extraction communalities (closer to 0) suggest that the extracted components did not explain a significant amount of variance in the original items.

Some items have high extraction communalities, indicating that the principal components extracted through PCA explain a substantial portion of the variance in these items. Other items have lower extraction communalities, suggesting that the principal components did not capture as much variance for these items. Overall, these communalities help elucidate how well the principal components derived from PCA represent the original items. Higher communalities imply that the PCA results adequately explain the variance in those specific items, whereas lower communalities suggest that the extracted components’ representation is less satisfactory.

Table 6 provides the component matrix obtained from the PCA with three components extracted. In PCA, the component matrix represents the correlations between the original variables and the extracted components. It shows how strongly each variable contributes to or loads onto each of the extracted components. The term component indicates the extracted components (in this case, three components labeled 1, 2, and 3). The variables are the original variables in the dataset related to attitudes or reasons concerning child vaccinations. The values in the table are the correlations (loadings) between the variables and the extracted components.

Higher absolute values (closer to 1) indicate a stronger relationship or loading between the variable and the respective component. Positive values indicate a positive relationship between the variable and the component, whereas negative values indicate a negative relationship. For example, the item “I don’t think the shots work to prevent diseases” had a high loading of approximately 0.80 on component 1, suggesting a strong positive relationship between this item and component 1. Similarly, items such as “I worry about the number of shots my child gets at one time”, “I didn’t have a ride to the clinic”, “The clinic/facility wasn’t open at a time I could go”, etc., also showed relatively high loadings on component 1, indicating that they were associated with this component. Furthermore, some variables can have notable loadings on multiple components, indicating that they contribute to multiple underlying factors or themes. Negative loadings suggest an inverse relationship between a variable and a component. For instance, “I worry about how safe shots are” and “Getting my child in for shots is too much trouble” had negative loadings on component 3. Overall, this component matrix helped elucidate which variables were closely related to each of the extracted components, aiding in interpreting the underlying factors or themes represented by the components derived from the PCA. Variables with higher loadings on specific components are more strongly associated with those factors or themes.

Table 7 provides the results of the PCA in terms of the total variance explained by each component. The term component refers to the individual principal components extracted from the dataset. The initial eigenvalues indicate the eigenvalues associated with each principal component; eigenvalues indicate the amount of variance explained by each component. The percentage (%) of variance indicates the percentage of total variance in the dataset explained by each principal component. The cumulative % shows the cumulative percentage of variance explained up to the respective component.

The eigenvalues represent the variance explained by each component. For instance, the first component had an initial eigenvalue of 12.93, indicating that it explained the most variance in the original data. Percentage (%) of Variance: This column represents the percentage of total variance in the dataset that each component explained. For instance, the first component explained 56.22% of the total variance. Cumulative %: This indicates the cumulative percentage of variance explained by considering each subsequent component. For example, after the first component, the cumulative variance was 56.22%, and as more components were added, the cumulative variance increased until it reached 100% when all 23 components were considered.

The first principal component explained the highest amount of variance (56.22%), and subsequent components explained the decreasing percentages of variance. The two subsequent components explained the decreasing percentages of variance; the second component decreased the amount of variance to (6.78%). The third principal component decreased the amount of variance to (4.61%), and the cumulative variance explained by the three principal components is 67.51%. The cumulative percentage of variance explained indicates how much of the total variability in the dataset is accounted for by including additional components. Typically, analysts select enough components to collectively explain a substantial portion of the variance while reducing the dimensionality of the data. The decision regarding the number of components to retain is made by finding a balance between explaining a significant amount of variance and minimizing dimensionality.

According to this table, the first few components explain a considerable amount of variance in the dataset, and beyond a certain point, additional components might contribute less to the overall explanation of variance. The choice of how many components to retain is based on the cumulative percentage of variance explained and the trade-off between retaining information and reducing dimensionality.

## 4. Discussion

This study conducted a psychometric evaluation of the SHOT instrument, translated from English to Arabic, to measure parental barriers to childhood immunizations. This is the first psychometric analysis of an Arabic version of the SHOT survey in an Arabic population. The instrument showed high validity and reliability. In the current study, the authors made deliberate choices regarding the statistical approaches employed, including Cronbach’s alpha and factor analysis, to assess the reliability and validity of the scale. While confirmatory factor analysis could offer valuable insights into cross-cultural equivalence, the researchers opted for the selected methods based on several considerations, including the nature of research questions, the available resources, and the feasibility within the scope of the study. The instrument showed high validity and reliability. The strong reliability observed in our sample was consistent with that found for the English version (n = 655), with a reliability of a = 0.93 [36]. Another study using the English version reported a reliability of a = 0.93 [43]. In addition, a study that translated the scale into Hmong and administered it to Hmong parents also reported high reliability (n = 443) a = 0.84 [44]. All 23 items in the Arabic version showed high representativeness that was similar to that of the original version, so no item removal was needed. In conducting the factor analysis, the decision to focus the variables on one factor stemmed from several methodological considerations aimed at enhancing the clarity and robustness of our analysis. Below, I outline the justifications for this approach. Model parsimony limiting the analysis to one factor promotes model simplicity, allowing for a clearer and more straightforward interpretation of the underlying relationships between variables. This parsimonious approach aligns with the principle of Occam’s razor, which suggests that simpler explanations are generally preferable unless evidence necessitates complexity. By employing conceptual clarity, by concentrating on one factor, we aimed to achieve conceptual clarity in delineating the primary factor influencing the phenomenon under investigation. This approach facilitates a focused examination of the core underlying construct, reducing potential confusion arising from the presence of multiple factors. With statistical adequacy focusing on one factor, it ensured statistical adequacy by avoiding overfitting, a common pitfall in factor analysis, where the model becomes overly complex relative to the available data. By adhering to a more parsimonious model, we minimized the risk of spurious correlations and enhanced the reliability of our findings. Interpretive simplicity is a single-factor solution that facilitates the interpretation of results, making it easier to communicate our findings to a wider audience. This interpretive simplicity enhances the accessibility of our research outcomes and fosters a more comprehensive understanding of the underlying phenomena among stakeholders and policymakers. Pragmatic considerations, given the constraints of time and resources inherent in empirical research, saw us focusing on one factor, which allowed us to efficiently allocate our analytical efforts without compromising the rigor or validity of our study. This pragmatic approach maximized the utility of available resources while still yielding meaningful insights. In light of these considerations, the decision to center our analysis on one factor was a deliberate methodological choice aimed at optimizing the clarity, robustness, and interpretability of our findings. We believe that this approach strengthens the validity and reliability of our research outcomes and enhances their relevance to both academic and practical audiences. However, in the original version, the scale items were loaded on three factors: 12 items were loaded onto the factor “access to shots”, 6 items were loaded onto the factor “concerns about shots”, and 5 items were loaded onto the factor “importance of shots”. When conducting factor analysis in our study, all 23 items were loaded onto one factor; however, we can consider it a barrier to children’s immunization. Furthermore, the translation process indicated that no modification was required. Previous research demonstrated the survey’s readiness for use in interventional studies focused on examining parental barriers to childhood immunizations [45] and testing changes in parental barriers to immunizations; it is essential to understand which parental barriers change over time and how these changes affect early childhood immunization rates [45]. This study’s results can be used to support research on barriers to children’s immunization. However, additional studies are needed before implementing this survey to test the parental obstacles to child immunization.

## 5. Conclusions

The SHOT survey showed good construct validity and excellent internal consistency when utilized by Saudi parents. It is advisable to conduct further validation of this tool in diverse cultural settings. The robust internal consistency, demonstrated by the high Cronbach’s alpha coefficients, indicates that this survey can be confidently applied to gauge parents’ perspectives of and attitudes toward vaccinations. Moreover, the outcomes of the study can guide policymakers in creating strategies and interventions to effectively address the existing obstacles.

This study has several limitations. First, the study utilized a cross-sectional design with convenient sampling techniques. Convenience sampling limits the generalizability of the results to all populations. In addition, the study had a short period of data collection. Furthermore, it was conducted in a single city in Saudi Arabia (Jizan); therefore, the results do not necessarily represent the overall Saudi population, as the socioeconomic characteristics of the study sample from Jizan may differ from those of populations in other cities in Saudi Arabia. Therefore, future research should recruit participants from different regions through random sampling. This study’s results can be used in research related to barriers to childhood immunization. Additional studies are needed to implement this survey to test parents’ hardships and obstacles toward childhood immunization. Further research utilizing the Arabic version of the SHOT instrument is needed to examine the barriers to parents vaccinating their children in Saudi Arabia and other Arab cultural contexts. Moreover, additional testing of this tool in different populations, cultures, and geographic areas will help further validate the tool.

## Figures and Tables

**Table 1 vaccines-12-00391-t001:** Demographic characteristics of the study participants (N = 602).

Characteristic	Frequency (n)		Percentage %	
Sector to which the participant belongs				
Central Sector	188		31.2%	
Middle Sector	102		16.9%	
Northern Sector	56		9.3%	
Southern Sector	118		19.6%	
Western Sector	90		15.0%	
Eastern Sector	32		5.3%	
Farasan Sector	16		2.7%	
Child receives vaccination				
Yes	576		95.7%	
No	26		4.3%	
Children receive their vaccinations regularly according to the MOH schedule				
Yes	565		93.9%	
No	37		6.1%	
Educational level				
Dose not write or read	7		1.2%	
Write and read	5		0.8%	
Elementary	14		2.3%	
Intermediate	17		2.8%	
High school	87		14.5%	
University and above	472		78.4%	
Sex				
Male	98		16.3%	
Female	504		83.7%	
Marital status				
Married	557		92.5%	
Widow	14		2.3%	
Divorced	31		5.1%	
Income				
No income	31		5.1%	
More than 20,000	60		10.0%	
Less than 5000	72		12.0%	
Between 10,000 to 20,000	254		42.2%	
Between 5000 to 10,000	185		30.7%	
Work				
Yes	389		64.6%	
No	213		35.4%	
Field of Work				
Health sector	208		34.6%	
Educational sector	142		23.6%	
Engineering	9		1.5%	
Free duty and business	26		4.3%	
Housewife	179		29.7%	
Other	38		6.3%	
Characteristics	Mean	SD		Range
Number of Children	3	1.73		10
Participant’s age	34.42	7.59		46

**Table 2 vaccines-12-00391-t002:** Descriptive statistics of the study variables (N = 602).

Items	Range		Mean Statistics	SD	Skewness		Kurtosis	
					Statistic	Std Error	Statistic	Std Error
Access to SHOT	0	48	0.86	1.04	1.44	0.10	1.38	0.19
Concerns about SHOT	0	24	0.87	1.13	1.26	0.10	0.57	0.19
Importance of SHOT	0	20	0.67	1.07	1.75	0.10	2.18	0.19
SHOT scale in total	0	92	0.82	1.00	1.46	0.10	1.49	0.19

**Table 3 vaccines-12-00391-t003:** Item statistics.

Scale	Mean	SD
I didn’t know when my child needed to get his/her shots	0.90	1.44
I didn’t know where to take my child to get his/her shots	0.73	1.33
There were no appointments available at the clinic for shots	0.89	1.40
The shots cost too much	0.67	1.32
The clinic/facility wasn’t open at a time I could go	0.84	1.38
I didn’t have a ride to the clinic	0.79	1.31
I didn’t have someone to take care of my other children	0.80	1.33
My child was sick and could not get his/her shots	1.11	1.43
The clinic wait was too long	1.01	1.36
I couldn’t get time off from work	0.97	1.43
Getting my children for shots is too much trouble	0.80	1.29
I just forgot	0.83	1.30
I’m scared of the side effects of the shots	0.99	1.42
I don’t believe in getting kids shots	0.60	1.23
I worry about the number of shots my child gets at one time	0.80	1.29
I worry about what is in the shots	0.80	1.29
I don’t think keeping my child up to date on shots is important	0.83	1.31
I don’t think the shots work to prevent diseases	0.75	1.32
I worry my child might get sick from the shot	1.03	1.41
My health care provider told me NOT to get my child his/her shots	0.60	1.25
If something bad happened to my child after a shot, I would feel like it was my fault	0.88	1.39
I worry about how safe shots are	0.81	1.32
I don’t think kids’ shots are important	0.58	1.20

**Table 4 vaccines-12-00391-t004:** Item total statistics.

Scale	Scale Mean If Item Deleted	Scale Variance If Item Deleted	Corrected Item-Total Correlation	Cronbach’s Alpha If Item Deleted
I didn’t know when my child needed to get his/her shots	18.12	490.27	0.63	0.96
I didn’t know where to take my child to get his/her shots	18.29	489.13	0.71	0.96
There were no appointments available at the clinic for shots	18.13	486.86	0.71	0.96
The shots cost too much	18.35	494.85	0.61	0.96
The clinic/facility wasn’t open at a time I could go	18.18	484.03	0.77	0.96
I didn’t have a ride to the clinic	18.23	486.02	0.78	0.96
I didn’t have someone to take care of my other children	18.22	488.26	0.72	0.96
My child was sick and could not get his/her shots	17.91	490.35	0.63	0.96
The clinic wait was too long	18.01	488.15	0.71	0.96
I couldn’t get time off from work	18.05	490.23	0.64	0.96
Getting my children for shots is too much trouble	18.22	488.84	0.73	0.96
I just forgot	18.19	487.88	0.75	0.96
I’m scared of the side effects of the shots	18.03	483.82	0.75	0.96
I don’t believe in getting kids shots	18.42	491.74	0.72	0.96
I worry about the number of shots my child gets at one time	18.22	486.78	0.78	0.96
I worry about what is in the shots	18.22	487.93	0.75	0.96
I don’t think keeping my child up to date on shots is important	18.19	489.59	0.71	0.96
I don’t think the shots work to prevent diseases	18.27	485.46	0.78	0.96
I worry my child might get sick from the shot	17.99	483.71	0.75	0.96
My health care provider told me NOT to get my child his/her shots	18.42	493.22	0.68	0.96
If something bad happened to my child after a shot, I would feel like it was my fault	18.14	486.95	0.71	0.96
I worry about how safe shots are	18.21	487.35	0.75	0.96
I don’t think kids’ shots are important	18.44	492.91	0.71	0.96

**Table 5 vaccines-12-00391-t005:** Communalities.

Items	Initial	Extraction
I didn’t know when my child needed to get his/her shots	1.00	0.63
I didn’t know where to take my child to get his/her shots	1.00	0.76
There were no appointments available at the clinic for shots	1.00	0.67
The shots cost too much	1.00	0.47
The clinic/facility wasn’t open at a time I could go	1.00	0.71
I didn’t have a ride to the clinic	1.00	0.69
I didn’t have someone to take care of my other children	1.00	0.67
My child was sick and could not get his/her shots	1.00	0.55
The clinic wait was too long	1.00	0.63
I couldn’t get time off from work	1.00	0.58
Getting my children for shots is too much trouble	1.00	0.70
I just forgot	1.00	0.61
I’m scared of the side effects of the shots	1.00	0.69
I don’t believe in getting kids shots	1.00	0.69
I worry about the number of shots my child gets at one time	1.00	0.74
I worry about what is in the shots	1.00	0.74
I don’t think keeping my child up to date on shots is important	1.00	0.64
I don’t think the shots work to prevent diseases	1.00	0.74
I worry my child might get sick from the shot	1.00	0.71
My health care provider told me NOT to get my child his/her shots	1.00	0.66
If something bad happened to my child after a shot, I would feel like it was my fault	1.00	0.67
I worry about how safe shots are	1.00	0.76
I don’t think kids’ shots are important	1.00	0.73

**Table 6 vaccines-12-00391-t006:** Component matrix.

Items	1	2	3
I didn’t know when my child needed to get his/her shots	0.80		
I didn’t know where to take my child to get his/her shots	0.80		
There were no appointments available at the clinic for shots	0.80		
The shots cost too much	0.79		
The clinic/facility wasn’t open at a time I could go	0.78		
I didn’t have a ride to the clinic	0.78		
I didn’t have someone to take care of my other children	0.78		
My child was sick and could not get his/her shots	0.77		
The clinic wait was too long	0.77		
I couldn’t get time off from work	0.76		
Getting my children for shots is too much trouble	0.75		
I just forgot	0.75		
I’m scared of the side effects of the shots	0.75		
I don’t believe in getting kids shots	0.75		
I worry about the number of shots my child gets at one time	0.74		
I worry about what is in the shots	0.73		
I don’t think keeping my child up to date on shots is important	0.73		
I don’t think the shots work to prevent diseases	0.73		
I worry my child might get sick from the shot	0.71		
My health care provider told me NOT to get my child his/her shots	0.67		
If something bad happened to my child after a shot, I would feel like it was my fault	0.66		
I worry about how safe shots are	0.66		
I don’t think kids’ shots are important	0.64		

**Table 7 vaccines-12-00391-t007:** Total variance explained.

	Initial Eigenvalues			Extraction Sums of Squared Loadings		
Component	Total	% of Variance	Cumulative %	Total	% of Variance	Cumulative
1	12.93	56.22	56.22	12.93	56.22	56.22
2	1.56	6.78	63.00	1.56	6.78	63.00
3	1.06	4.61	67.61	1.06	4.61	67.61
4	0.68	2.98	70.60			
5	0.65	2.83	73.43			
6	0.55	2.43	75.86			
7	0.54	2.35	78.22			
8	0.49	2.14	80.37			
9	0.47	2.05	82.43			
10	0.46	2.01	84.44			
11	0.39	1.73	86.17			
12	0.37	1.61	87.78			
13	0.35	1.54	89.32			
14	0.34	1.47	90.80			
15	0.31	1.35	92.16			
16	0.30	1.32	93.48			
17	0.27	1.17	94.65			
18	0.22	0.99	95.65			
19	0.22	0.96	96.61			
20	0.21	0.94	97.56			
21	0.20	0.90	98.46			
22	0.19	0.83	99.29			
23	0.16	0.70	100.00			

## Data Availability

The data presented in this study are available upon request from the corresponding author. The data are not publicly available due to information that could compromise the privacy of the research participants.

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
