# Peer review of "A Psychometric Study of the Arabic Version of the “Searching for Hardships and Obstacles to Shots (SHOT)” Instrument for Use in Saudi Arabia"

_vaccines, 2024, doi:10.3390/vaccines12040391_

Round 1

Reviewer 1 Report

Comments and Suggestions for Authors

Thank you for choosing me as a reviewer for this manuscript entitled ‘A Psychometric Study of the Arabic Version of the “Searching 2 for Hardships and Obstacles to Shots” Instrument for Use in 3 Saudi Arabia.’  The manuscript is befitting the aim and scope of the Vaccine’s journal under the theme of  ‘Vaccine Acceptance/Hesitancy’. Overall, the manuscript is coherent, and has the key stipulated sections Abstract, Introduction, Methods, and Discussions. However, it needs several edits, major revisions, and edits that are suggested below, for being considered for publication. Advised language and punctuation revisions and suggested edits are also mentioned as track-changed comments in the attached manuscript review document.

Title:

Please consider adding the acronym (SHOTS) after the full form Searching 2 for Hardships and Obstacles to Shots”.

Authors

If  the first author has two affiliations 1a, and 1b, both numbers should be superscripted. Please delete the space between Hobani and 1, and the ‘,’ after author 2.

Abstract:

Reads fine now, but might need revisions after the article is revised

Keywords:
Authors can consider adding ‘Psychometric tools for vaccine hesitancies’

Introduction:

Several sentences do not have supporting citations. Also, the introduction section is too long highlighting only parental hesitancies to childhood vaccines. Authors may consider 1. Synopsizing the section, and 2. Arranging and rewriting it in a way that addresses three aspects – (i) parental perceptions and behaviors that have led to negative vaccination decisions by the former for their/ others’ children, (ii) examples of utilization of culturally sensitive and translated tools for better identification of vaccine perceptions among community members (this part is completely missing)

Some suggested references

·       Domek, G. J., O'Leary, S. T., Bull, S., Bronsert, M., Contreras-Roldan, I. L., Ventura, G. A. B., ... & Asturias, E. J. (2018). Measuring vaccine hesitancy: Field testing the WHO SAGE Working Group on Vaccine Hesitancy survey tool in Guatemala. Vaccine, 36(35), 5273-5281.

·       Oduwole, E. O., Pienaar, E. D., Mahomed, H., & Wiysonge, C. S. (2019). Current tools available for investigating vaccine hesitancy: a scoping review protocol. BMJ open, 9(12), e033245.

·       Dutta, T., Agley, J., Lin, H. C., & Xiao, Y. (2021, May). Gender-responsive language in the National Policy Guidelines for Immunization in Kenya and changes in prevalence of tetanus vaccination among women, 2008–09 to 2014: A mixed methods study. In Women's Studies International Forum (Vol. 86, p. 102476). Pergamon.

·       Howard, M. C. (2022). A more comprehensive measure of vaccine hesitancy: Creation of the Multidimensional Vaccine Hesitancy Scale (MVHS). Journal of Health Psychology, 27(10), 2402-2419.

·       Alsubaie, S. S., Gosadi, I. M., Alsaadi, B. M., Albacker, N. B., Bawazir, M. A., Bin-Daud, N., ... & Alzamil, F. A. (2019). Vaccine hesitancy among Saudi parents and its determinants: Result from the WHO SAGE working group on vaccine hesitancy survey tool. Saudi medical journal, 40(12), 1242.

Methods:

For better clarity, I’d suggest having three subsections describing the following (i) the setting and vaccination governance system in the study area, (ii) an introduction and summary of the SHOT tool in English and thereafter describing the translation and back translation processes, and (iii) the application and testing of the SHOTS tool among the community. Authors may also add a flowchart with timelines to elucidate the process.

If IRB was received that needs to be mentioned. There needs to be clarity whether permission in writing was sought to undertake the translation, and informed of implied consent was sought, for administering the tool among community members.

How many participants were sought per PHC?

Results: Results from both the sections needs to be mentioned in separate sub-sections- the translation and back translation and testing of the SHOTS tool among the community.

Discussion

This needs to critically discuss the aspects highlighted in the Results section. The Discussion section is brief and has some parts which are better fitted to the Results section.

Conclusions: Any implementation recommendations that the authors would like to add?

Comments on the Quality of English Language

Author Response

Dear Reviewer

Suggested modification attached in the file 

Thank You

Reviewer 2 Report

Comments and Suggestions for Authors

            This study showed that results will help policymakers develop programs or interventional initiatives to overcome these barriers.

           Authors are kindly requested to emphasize the current concepts about these issues in the context of recent knowledge and the available literature. This articles should be quoted in the References list.

Comments on the Quality of English Language

Good

Author Response

Dear Reviewer

Suggested modificationns attache in thhe file

Thank You

Reviewer 3 Report

Comments and Suggestions for Authors

This paper describes the translation of the “Searching for Hardships and Obstacles to Shots” (SHOTS) instrument from English to Arabic and the statistical analysis and findings from a psychometric evaluation of the Arabic version to measure parental barriers to childhood immunization.

The paper generally is well written and the study follow standard research methods for such a study.

However, there is one item in the Abstract (and also in lines 423-426 of the text) that needs attention. This is the apparent contradiction between the sentence "The first principal component explained the highest variance (56.22%), and subsequent components explained decreasing percentages of variance." and the next sentence that states: "The third principal component explained the highest variance (67.61%), and subsequent components explained decreasing percentages of variance." Actually, in Table 7 it is reported that the third principal component explained 4.61% of the variance, and the cumulative variance explained by the three principal components is 67.51%. This needs to be corrected in the Abstract and the text. 

Author Response

(The authors gave the same response as above.)

Reviewer 4 Report

Comments and Suggestions for Authors

Dear Editor, Dear Authors,

It was a pleasure to read and make a revision of the manuscript entitled ‘A Psychometric Study of the Arabic Version of the “Searching 2 for Hardships and Obstacles to Shots” Instrument for Use in 3 Saudi Arabia’.

The content is well written, sound and logic, however there are some issues which should be considered before publication:

*abstract: “The third principal component explained the highest variance (67.61%)” is mistaken …67.61% represents the cumulative variance, meaning the first three components explained 67.61% … the third component explained additionally 4.61% (to the two previous components). The same inconsistency is on page 14 – line 423-426, correct it please;

*p.2-l.72: the form of citation in the manuscript does not follow the required standard;

*p.2-l.80-81: the message provided by the sentence is like 100% of children in Saudi Arabia is vaccinated, but probably it is not … and the proportions reflect the part which is vaccinated;

*Regarding the cultural adaptation of the scale … in my opinion the final translated version (in Arabic) should be presented in a supplement/appendix;

*The content of Table 2 is hard to understand. The Tab. title says ‘descriptive … (sum of total)’ and next e.g. ‘access to SHOT’ mean 0.86 … so how the reader should interpret / understand the value of 0.86? … It should be clarified. Additionally, I suggest adding min & max

*why the name of the tool has been changed from SHOTS to SHOT?

*Table 3: there are means with SD, but it is hard to understand those values without knowledge how the options from the 4-point Likert scale were coded. Add under methods the values representing answer options used in the scale;

*provide a note for PCA, please, whether you used any rotations or not, and some information on how well the model fits to the data (some model fit parameters);

*p.14-l.435: this is not exactly so … there are several methods to extract the number of factors. These include the Scree Test, the parallel analysis, the Velicer's Minimum Average Partial Test, the very simple structure method and the eigenvalues criterion. And e.g. eigenvalues criterion (other results are not presented in the manuscript) shows 3 component solution, but not one as mentioned by authors. Therefore the decision/comments on ‘one factor solution’ should be clearly justified as the results provided do not support it.

Best wishes,

Reviewer

Comments on the Quality of English Language

There are some minor issues with English grammar, like “The SHOTS survey showed good construct validity and excellent internal consistency when utilized by Saudi parents” … I am not sure whether the SHOTS was used by Saudi parents, or to assess hardships and obstacles to shots in Saudi parents. I suggest reviewing the manuscript.

Author Response

(The authors gave the same response as above.)

Reviewer 5 Report

Comments and Suggestions for Authors

This paper is an interesting study and is excellent from a methodological point of view.

Here are some comments that may improve the manuscript.

1.      In the introduction write about the effect of covid-19 in vaccine coverage, and also the effect that vaccine fatigue can have on vaccinations. There is an extense bibliograpy on the topic.

2.      Review line 256-257 “Also, 92.5% of participant were married (n=557), 5.1% (n= 256 14) were divorced, and only 5.1% (n= 31) were divorced. “ some thing is wrong here. How many are divorced 14 or 31?

3.      The study is very well designed, the authors do power computations. They computed cronbach alpha, Factor Analysis. Ofte when a scale is translate to another language a posible statistical analysis is to do correspondence factor analysis (CFA) allow to compare the structure of the factors in the new language with that in the original language.

4.      It is true that the computations may be hard to do.  The authors should explain in the dicussion why the didn’t choose to compute CFA, and used the aproach used in the manuscript.

5.      The authors use the word gender: Gender is related with the sexual orientation, sex is related with biological charasteristic. Sex - refers to biological differences between females and males, including chromosomes, sex organs, and endogenous hormonal profiles.Gender-  refers to socially constructed and enacted roles and behaviors which occur in a historical and cultural context and vary across societies and over time. Please see Sex & Gender | Office of Research on Women's Health (nih.gov)  https://orwh.od.nih.gov/sex-gender

 6. Include the translation of the questionnaire into Arabic as an annex file. 

Author Response

(The authors gave the same response as above.)

Round 2

Reviewer 1 Report

Comments and Suggestions for Authors

The authors have mostly incorporated all the review comments. I would suggest a thorough read to improve the language and flow. Especially in the Introduction section, more needs to be added about psychometric instruments to measure vaccine hesitancies, and attitudes about other biomedical tools, while reducing the text on vaccine hesitancy per se. Similarly, more needs to be added on some of the social and political determinants of health that are associated with vaccine hesitancies among specific populations and how a translated tool/ communication could be hypothesized as a more effective outreach method and process. For, the later part (how locally responsive, and culturally resonating tools/ programs are better situated) more references are suggested. 

Balgiu, B. A., Sfeatcu, R., Țâncu, A. M. C., Imre, M., Petre, A., & Tribus, L. (2022). The multidimensional vaccine hesitancy scale: A validation study. Vaccines10(10), 1755.

Dutta, T., Agley, J., Lin, H. C., & Xiao, Y. (2021, May). Gender-responsive language in the National Policy Guidelines for Immunization in Kenya and changes in prevalence of tetanus vaccination among women, 2008–09 to 2014: A mixed methods study. In Women's Studies International Forum (Vol. 86, p. 102476). Pergamon.

Dambi, J. M., Corten, L., Chiwaridzo, M., Jack, H., Mlambo, T., & Jelsma, J. (2018). A systematic review of the psychometric properties of the cross-cultural translations and adaptations of the Multidimensional Perceived Social Support Scale (MSPSS). Health and quality of life outcomes16, 1-19.

Merhi, R., & Kazarian, S. S. (2012). Validation of the Arabic translation of the Multidimensional Scale of Perceived Social Support (Arabic-MSPSS) in a Lebanese community sample. Arab Journal of Psychiatry23(2), 159-168.

Comments on the Quality of English Language

The English is fine 

Author Response

Dear reviewer

Thank you for your comments, which will help us execute an excellent publication.

I would appreciate your comment.

attached document for changes made. 

Thank You
